# Novel Species and Records of *Dictyosporiaceae* from Freshwater Habitats in China and Thailand

**DOI:** 10.3390/jof8111200

**Published:** 2022-11-14

**Authors:** Hongwei Shen, Danfeng Bao, Dhanushka N. Wanasinghe, Saranyaphat Boonmee, Jiankui Liu, Zonglong Luo

**Affiliations:** 1College of Agriculture and Biological Science, Dali University, Dali 671003, China; 2School of Science, Mae Fah Luang University, Chiang Rai 57100, Thailand; 3Center of Excellence in Fungal Research, Mae Fah Luang University, Chiang Rai 57100, Thailand; 4Department of Entomology and Plant Pathology, Faculty of Agriculture, Chiang Mai University, Chiang Mai 50200, Thailand; 5Honghe Center for Mountain Futures, Kunming Institute of Botany, Chinese Academy of Sciences, Honghe 654400, China; 6Center for Informational Biology, School of Life Science and Technology, University of Electronic Science and Technology of China, Chengdu 611731, China

**Keywords:** new species, asexual morph, lignicolous freshwater fungi, phylogeny, taxonomy

## Abstract

China and Thailand are rich in fungal diversity with abundant freshwater resources that are favorable for numerous fungal encounters. Resulting from this, the majority of the *Dictyosporiaceae* species reported were from these two countries. During the investigation on the diversity of lignicolous freshwater fungi in the Greater Mekong Subregion, eleven collections of cheirosporous species on submerged wood were collected from lentic and lotic habitats in China and Thailand. Phylogenetic analysis that combined nuclear small-subunit ribosomal RNA (SSU), internal transcribed spacer region (ITS), nuclear large subunit ribosomal RNA (LSU) and translation elongation factor 1α (*tef 1-α*) loci revealed six new species: *Dictyocheirospora chiangmaiensis*, *D. multiappendiculata*, *D. suae*, *Digitodesmium aquaticum*, *Vikalpa grandispora* and *V. sphaerica.* In addition, four known species were also identified and reported based on morphological and phylogenetic evidence. The detailed descriptions and illustrations of these taxa are provided with an updated phylogenetic tree of *Dictyosporiaceae*.

## 1. Introduction

Several studies investigating the ecology of aquatic fungi in early 2000 [1,2,3,4] recorded numerous species of cheiroid hyphomycetes, many in *Dictyosporium* and several other genera of *Dictyosporiaceae*. *Dictyosporiaceae* (*Pleosporales*) comprises 20 genera, the species of these genera are distributed worldwide and most taxa are saprobes on plant litter, especially dead or decaying wood in freshwater and terrestrial habitats [5,6,7,8,9,10,11,12,13,14]. Most of the asexual morphic members of the *Dictyosporiaceae* are hyphomycetous and characterized by the production of cheiroid, digitate, palmate and/or dictyosporous, and pale brown to brown conidia, viz., *Aquadictyospora* [8], *Aquaticheirospora* [7], *Cheirosporium* [6], *Dictyocheirospora* [5], *Dictyopalmispora* [5], *Dictyosporium* [15], *Digitodesmium* [16], *Jalapriya* [5], *Neodigitodesmium* [12], *Pseudodictyosporium* [17] and *Vikalpa* [5]. A few genera have coelomycetous asexual morphs, viz., *Immotthia* [18], *Pseudocoleophoma* [19], *Pseudoconiothyrium* [20], *Pseudocyclothyriella* [11], *Sajamaea* [21] and *Verrucoccum* [14]. Sexual morphs are known only from four genera *Dyctiosporium* [15], *Gregarithecium* [19], *Immotthia* [18], *Pseudocoleophoma* [19] and *Verrucoccum* [14].

*Dictyosporium* typified by *D. elegans* Corda [15] is characterized by effuse or compact granular colonies; micronematous, mononematous and branched conidiophores; monoblastic, terminal, determinate, doliiform, spherical or subspherical conidiogenous cells; and cheiroid, solitary, branched and complanate conidia, with or without appendages, comprising multiple rows of cells. The sexual morphs are characterized by subglobose ascomata, cylindrical asci and hyaline, fusiform, uniseptate ascospores, with or without a sheath [5,9,10,22]. *Dictyosporium* was placed in *Pleosporales* based on a phylogenetic analysis of LSU/SSU [23,24]. Liu et al. [25], Tanaka et al. [19] and Boonmee et al. [5] determined the phylogenetic position of *Dictyosporium* in *Dictyosporiaceae* (*Pleosporales*) based on multi-gene phylogeny analysis. Prasher and Verma [26] and Silva et al. [27] provided an account of *Dictyosporium* species with profiles and comparisons, including 48 accepted species. Since this publication, ten further species were recorded in this genus, and 13 species were relocated to *Dictyocheirospora*, *Jalapriya* and *Vikalpa* [5,9,10,19,26,28,29,30,31,32,33]. Currently, 54 species are accepted in *Dictyosporium*, of which, 17 were collected from the freshwater habitat, while 37 were collected from terrestrial habitats.

*Dictyocheirospora* was established by Boonmee et al. [5] to accommodate species with dark sporodochial colonies that produce aeroaquatic cheiroid, non-complanate or cylindrical conidia, mostly with conidial arms that are closely gathered together at the apex, and with *Dictyocheirospora* rotunda as the type species. Boonmee et al. [5] introduced three new species and transferred four species of *Dictyosporium* to *Dictyocheirospora* based on phylogenetic results. The difference between *Dictyocheirospora* and *Dictyosporium* is that the former’s conidia arms are on different planes and closely located at the apex, while in the latter, the conidia arms are closely arranged on the same plane. Based on these two morphological features, members of *Dictyocheirospora* and *Dictyosporium* are easily distinguished. Therefore, based on morphological and phylogenetic analysis, 23 species are accepted in *Dictyocheirospora*, including seven species that were transferred from *Dictyosporium* and one species from *Cattanea* [5,9,10,34,35,36,37,38,39,40].

*Digitodesmium* was introduced by Kirk [16] to accommodate *D. elegans* P.M. Kirk and it is characterized by the euseptate, cheiroid (digitate) conidia produced in sporodochia, the arms of conidia scattered in different planes and an apical gelatinous cap [12,16,35,41]. Currently, nine epithets of *Digitodesmium* are listed in Index Fungorum (accessed on 22 September 2022), except for *Digitodesmium bambusicola* L. Cai, K.Q. Zhang, McKenzie, W.H. Ho & K.D. Hyde; *D. chiangmaiense* Q.J. Shang & K.D. Hyde; *D. polybrachiatum* T.F. Nóbrega, B.W. Ferreira & R.W. Barreto; and *D. tectonae* (Rajeshk., Rajn. K. Verma, Boonmee, K.D. Hyde, Chandrasiri & Wijayaw.) W.H. Tian & Maharachch., where the remaining five species lack molecular sequence data [12,16,41,42,43,44]. Members of *Digitodesmium* are mainly distributed in terrestrial and freshwater environments in Asia and Europe [20,35,41,42,43,45]. Two species were reported in China, with both from submerged wood in freshwater environments, and one in Thailand reported from terrestrial environments [35,41,45].

*Vikalpa* was established by Boonmee et al. [5] to accommodate species characterized by sporododochial conidiomata, septate conidia and three rows of cells in different planes. *Vikalpa* is a sparse group that contains only four species, of which *Vikalpa australiensis* (B. Sutton) D’souza, Boonmee & K.D. Hyde; *V. freycinetiae* (McKenzie) D’souza, Boonmee & K.D. Hyd; and *V. micronesiaca* (Matsush.) D’souza, Bhat & K.D. Hyde were collected from decaying wood and leaves in terrestrial habitats of Australia, New Zealand and Micronesia [46,47,48], and only *V. lignicola* D’souza, Bhat, H.Y. Su & K.D. Hyde was from a lotic freshwater habitat in Yunnan, China [5]. The sequence data (ITS) is only available for the type species.

The Greater Mekong Subregion includes Yunnan Province in China and Thailand and is a highly diverse region for plants and fungi [49,50,51,52,53]. We continuously studied the lignicolous freshwater fungi along a north/south gradient of the region [54] and eleven fresh collections were collected. The aim was to (1) clarify the phylogenetic positions and taxonomy of the eleven collections based on morphological and phylogenetic analysis; (2) provide a comparison of conidial morphological characteristics of *Dictyocheirospora*, *Digitodesmium* and *Vikalpa* species; and (3) provide insights into the lignicolous freshwater taxa and increase knowledge of microfungi in the Greater Mekong Subregion.

## 2. Materials and Methods

### 2.1. Specimen Collection, Examination and Isolation

The specimens of submerged decaying wood were collected from lentic (lakes; Fuxian, Erhai and Luguhu) and lotic (streams; Mae sai, Pong Pham and three unknown) habitats in China and Thailand. The samples were selected from wood substrates submerged in water with a diameter of 1–2 cm and a length of about 20 cm, including tree branches, twigs and rotten branches of grasses. The collected samples were placed in plastic ziplock bags and were taken to the laboratory for processing.

Morphological observations were done following Luo et al. [55] and Senanayake et al. [56] with a few modifications. The samples were incubated in a plastic box at room temperature for one week. Macromorphological characters of samples were observed using an Optec SZ 760 compound stereomicroscope. Temporarily prepared microscope slides were placed under a Nikon ECLIPSE Ni-U compound stereomicroscope for observation and micro-morphological photography. The morphologies of colonies on native substrates were photographed with a Nikon SMZ1000 stereo zoom microscope.

Single spore isolations were performed as follows: the tip of a sterile toothpick dipped in sterile water was used to capture the conidia of the target colony directly from the specimen; the conidia were then streaked on the surface of water agar (WA) or potato dextrose agar (PDA) and incubated at room temperature overnight. The single germinated conidia were transferred to fresh PDA plates and incubated at room temperature. A few of the remaining germinated spores in the media plate were separated along with agar by using a needle and transferred onto water-mounted glass slides for photographs to capture the germination position of the germ tubes.

After finalizing the observation and isolation, the specimens were dried under natural light, wrapped in absorbent paper and placed in a ziplock bag with mothballs. Specimens were deposited in the herbarium of Mae Fah Luang University (MFLU), Chiang Rai, Thailand, and the Kunming Institute of Botany, Chinese Academy of Sciences (KUN-HKAS), Kunming, China. The cultures were deposited with the Mae Fah Luang University Culture Collection (MFLUCC), China General Microbiological Culture Collection Center (CGMCC) and Kunming Institute of Botany Culture Collection (KUNCC). Facesoffungi numbers and MycoBank numbers were obtained as described in Jayasiri et al. [57] and MycoBank (https://www.mycobank.org; accessed on 8 October 2022).

### 2.2. DNA Extraction, PCR Amplification and Sequencing

DNA extraction, PCR amplification, sequencing and phylogenetic analysis were done in accordance with the methods of Dissanayake et al. [58]. Mycelia for DNA extraction from each isolate were grown on PDA for 3–4 weeks at room temperature. Total genomic DNA was extracted from 100–300 mg axenic mycelium via scraping from the edges of the growing culture using a sterile scalpel and transferred to a 1.5 mL microcentrifuge tube using sterilized inoculum needles. The mycelium was ground to a fine powder with liquid nitrogen or quartz sand to break the cells for DNA extraction. When the cultures could not be maintained with some of the collected samples, fruiting structures (20–50 mg) were removed from the natural substrate using a sterile scalpel and placed on sterile paper, and then transferred to a 1.5 mL microcentrifuge tube. DNA was extracted with the Trelief^TM^ Plant Genomic DNA Kit (TSP101) following the manufacturer’s guidelines.

Four gene regions, viz., ITS, LSU, SSU and *tef1-α*, were amplified using ITS5/ITS4 [59], LR0R/LR5 [60], NS1/NS4 [59] and EF1-983F/EF1-2218R [61] primer pairs, respectively. Primer sequences are available in the WASABI database on the AFTOL website (aftol.org). The PCR mixture contained 12.5 μL of 2 × GS Taq PCR MasterMix (mixture of DNA polymerase, dNTPs, Mg^2+^ and optimized buffer; Genesand Biotech, Beijing, China); 1 μL of each primer, including forward primer and reverse primer (10 μM); 1 μL template DNA extract; and 9.5 μL double-distilled water [55]. The PCR thermal cycling conditions of ITS and SSU were as follows: 94 °C for 3 min, followed by 35 cycles of denaturation at 94 °C for 30 s, annealing at 56 °C for 50 s, elongation at 72 °C for 1 min and a final extension at 72 °C for 10 min; the LSU and TEF1-α thermal cycling conditions were as follows: 94 °C for 3 min, followed by 35 cycles of denaturation at 94 °C for 30 s, annealing at 55 °C for 50 s, elongation at 72 °C for 1 min and a final extension at 72 °C for 10 min. PCR products were then purified using minicolumns, purification resin and buffer according to the manufacturer’s protocols. The sequences were carried out at the Beijing Tsingke Biological Engineering Technology and Services Co., Ltd. (Beijing, China).

### 2.3. Phylogenetic Analysis

BLAST searches using the BLASTn algorithm were performed to retrieve similar sequences from GenBank (http://www.ncbi.nlm.nih.gov, accessed on 05 May 2022) and relevant publications [10,40]. The sequences were aligned using the MAFFT online service: multiple alignment program MAFFT v.7 [62] (http://mafft.cbrc.jp/alignment/server/index.html, accessed on 11 May 2022), and sequence trimming was performed with trimAl v1.2 with default parameters (http://trimal.cgenomics.org (accessed on 11 May 2022) for specific operation steps) [63]. The sequence dataset was combined using SquenceMatrix v.1.7.8 [64]. FASTA alignment formats were changed to PHYLIP and NEXUS formats using the website Alignment Transformation Environment (ALTER) (http://sing.ei.uvigo.es/ALTER/, accessed on 21 September 2022).

Maximum likelihood (ML) analysis was performed by setting RAxML-HPC2 on XSEDE (8.2.12) [65,66] in the CIPRES Science Gateway [67] (http://www.phylo.org/portal2, accessed on 25 January 2022) using the GTR+GAMMA model with 1000 bootstrap repetitions. Bayesian analyses were performed in MrBayes 3.2.6 [68] and the best-fitting model of sequences evolution was estimated via MrModeltest 2.2 [69,70,71]. The Markov Chain Monte Carlo (MCMC) sampling approach was used to calculate posterior probabilities (PP) [72]. Bayesian analyses of six simultaneous Markov chains were run for 5 M generations and trees were sampled every thousand generations.

Phylogenetic trees were visualized using FigTree v1.4.0 (http://tree.bio.ed.ac.uk/software/figtree/ (accessed on 13 May 2022)), while editing and typesetting were achieved using Adobe Illustrator (AI) (Adobe Systems Inc., the United States). The new sequences were submitted in GenBank and the strain information used in this study is provided in Table 1.

## 3. Results

### 3.1. Phylogenetic Analysis

The dataset was composed of the combined SSU, ITS, LSU and *tef 1-α* sequence data of 124 taxa in *Dictyosporiaceae*, including 4041 characters (including gaps) with *Periconia igniaria* (CBS 379.86 and CBS 845.96) as the outgroup taxon (Figure 1). Maximum likelihood (ML) analysis and Bayesian analysis produced similar topologies that were consistent across the major clades. The likelihood of the final tree is evaluated and optimized using GAMMA. The best RAxML tree with a final likelihood value of −32,818.079790 is presented in Figure 1. The matrix had 1738 distinct alignment patterns, with 32.59% undetermined characters or gaps. The estimated base frequencies were as follows: A = 0.241143, C = 0.262390, G = 0.283430 and T = 0.213037; substitution rates AC = 1.312486, AG = 3.111299, AT = 1.426646, CG = 1.005458, CT = 6.740140, GT = 1.000000, α = 0.275558 and tree-length = 2.908192. The Bayesian analyses generated 1408 trees (average standard deviation of split frequencies: 0.009910) from which 1056 were sampled after 25% of the trees were discarded as burn-in. Bootstrap support values with an ML greater than 75%, and Bayesian posterior probabilities (PP) greater than 0.97 are given above the nodes.

The multigene phylogenetic analysis showed that eleven of our new strains were nested in *Dictyocheirospora*, *Dictyosporium*, *Digitodesmium* and *Vikalpa*. *Dictyocheirospora heptaspora* (MFLUCC 22-0096) was clustered with two strains of *D. heptaspora* (CBS 396.59 and DLUCC 1992), *D. aquadulcis* (MFLUCC 22-0095) and *D. nabanheensis* (MFLUCC 22-0094), and *Dictyosporium tubulatum* (KUN-HKAS 115789) was clustered with their ex-type strains. A new taxon, viz., *Dictyocheirospora chiangmaiensis* (MFLUCC 22-0097), formed a distinct clade with *D. clematidis* (MFLUCC 17-2089), *D. metroxylonis* (MFLUCC 15-0028a and MFLUCC 15-0028b), *D. taiwanense* (MFLUCC 17-2654) and *D. thailandica* (MFLUCC 18-0987). *Dictyocheirospora suae* (KUNCC 22-12424) and *D. multiappendiculata* (KUNCC 22-10734 and KUNCC 22-10736) aggregated to form a separate clade in the genus *Dictyocheirospora*. *Digitodesmium aquaticum* (MFLU 22-0203) clustered as a sister clade with *Di. bambusicola* (CBS 110279) and *Digitodesmium* sp. (TBRC 10037 and BRC 10038) with strong support (100% ML/1.00 PP). *Vikalpa grandispora* (KUNCC 22-12425) clustered with the new species *V. sphaerica* (CGMCC 3.20682) with low support (84% ML), and clustered as a sister clade with *V. australiensis* (HKUCC 10304).

### 3.2. Taxonomy

*Dictyocheirospora* M.J. D’souza, Boonmee & K.D. Hyde, Fungal Diversity 80: 465 (2016).

Notes: The current study found three new species in *Dictyocheirospora*. We also re-collected *Dictyocheirospora aquadulcis*, *D*. *heptaspora* and *D*. *nabanheensis* from freshwater habitats in Thailand. These taxa are subsequently illustrated and described below.

*Dictyocheirospora chiangmaiensis* H.W. Shen, S. Boonmee & Z.L. Luo sp. nov., Figure 2.

MycoBank number: MB 846309.

Etymology: “chiangmaiensis” refers to the Chiang Mai Province, Thailand, where the species was collected.

Holotype: MFLU 22-0199.

*Saprotrophic* on submerged decaying wood in freshwater habitats. Sexual morph: undetermined. Asexual morph: hyphomycetous. *Colonies* on a natural substrate were punctiform, sporodochial, scattered and brown. *Mycelium* was composed of immersed or partly superficial, pale brown, septate, smooth, thin-walled and branched hyphae. *Conidiophores* were micronematous, mononematous, septate, cylindrical, pale brown to brown, smooth and thin-walled. *Conidiogenous cells* were holoblastic, cylindrical, septate, hyaline to pale brown, smooth and thin-walled. *Conidia* were (40–)42–46(–48) × (–14)16–18(–20) μm ($\bar{x}$ = 44 × 17 μm, *n* = 30), solitary, cheiroid, ellipsoid to cylindrical, brown, not complanate, composed of 4–6 rows of cells, euseptate, unseparated, each row with 9–10 cells, slightly bent inward at the apex and sometimes with hyaline subglobose appendages at the subapical. *Conidial secession* was schizolytic.

Culture characteristics: Conidia germinating on PDA within 12 h and germ tubes produced at the base (Figure 2i). Colony growth was slow, reaching 2 cm after 8 weeks at room temperature. Mycelium was loose, flocculent, smooth-edged, yellow on the forward, middle mastoid, edge brown to brown-red and middle black on the reverse side.

Material examined: Thailand, Chiang Mai Province, around the Mushroom Research Center (MRC); 19°07′05″ N, 98°45′40″ E, (680 m); on submerged decaying wood in a stream; 9 July 2020; H.W. Shen; and SHW 22 (MFLU 22-0199, holotype) and ex-type culture (MFLUCC 22-0097).

Notes: Phylogenetically, *Dictyocheirospora chiangmaiensis* forms a distinct lineage basal to four *Dictyocheirospora* species, i.e., *D. clematidis* (MFLUCC 17-2089) [32], *D. metroxylonis* (MFLUCC 15-0028a, MFLUCC 15-0028b) [37], *D. taiwanense* (MFLUCC 17-2654) [35] and *D. thailandica* (MFLUCC 18-0987) [38] with good support (97% ML/1.00 PP, Figure 1). *Dictyocheirospora chiangmaiensis* can be distinguished from other species in this clade by its smaller conidia with fewer rows (4–6) of cells (Table 2).

*Dictyocheirospora suae* H.W. Shen & Z.L. Luo, sp. nov., Figure 3.

MycoBank number: MB 846010.

Etymology: “suae” (Lat.) in memory of the Chinese mycologist Prof. Hong-Yan Su (4 April 1967–3 May 2022).

Holotype: KUN-HKAS 121703.

*Saprotrophic* on submerged decaying wood in a freshwater lake. Sexual morph: undetermined. Asexual morph: hyphomycetous. *Colonies* on a natural substrate were punctiform, sporodochial, scattered and brown. *Mycelium* was composed of immersed or partly superficial, brown to dark brown, septate, smooth, thin-walled and branched hyphae. *Conidiophores* were micronematous, mononematous, septate, cylindrical, pale brown, smooth and thin-walled. *Conidiogenous cells* were holoblastic, cylindrical, septate, hyaline to pale brown, smooth, thin-walled and sometimes lacked conidiogenous cells. *Conidia* were (65–)72–79 × (–17)20–25(–29) μm ($\bar{x}$ = 76 × 23 μm, *n* = 30), solitary, cheiroid, ellipsoid to cylindrical, not complanate, brown, composed of 5–7 rows of cells, unseparated, euseptate, tightly clustered at the apex of rows, each row with 12–15 cells and with globose to subglobose apical appendages. *Conidial secession* was schizolytic.

Culture characteristics: Conidia germinated on PDA within 12 h and germ tubes were produced at the base of cells (Figure 3i). Colonies on PDA reached about 4 cm in one month at room temperature. Mycelium was loose, flocculent and white with flaxen spots on the forward.

Material examined: China, Yunnan Province, Dali City, Erhai Lake; 25°49′04″ N, 100°08′46″ E, (1790 m); on submerged decaying wood; 1 April 2021; S.P. Huang; and 3EH GSC 8-6-2 (KUN-HKAS 121703, holotype) and ex-type cultures (KUNCC 22-12424).

Notes: Phylogenetic analysis showed that *Dictyocheirospora suae* clustered as a sister taxon to *D. multiappendiculata* with good support (95% ML/1.00 PP, Figure 1). *Dictyocheirospora suae* and *D. multiappendiculata* shared similar morphology in having solitary, cheiroid, ellipsoid to cylindrical and not complanate conidia with apical appendages. However, *D. suae* had larger conidia (72–79 × 20–25 μm vs. 36–46 × 13–18 μm) and more cells in each row (12–15 vs. 9–13). In addition, the conidia of *D. multiappendiculata* had more appendages than *D. suae*. A comparison of the ITS sequences of *D. suae* and *D. multiappendiculata* showed 2.35% (11/469 bp) nucleotide differences.

*Dictyocheirospora multiappendiculata* H.W. Shen & Z.L. Luo, sp. nov., Figure 4.

MycoBank number: MB 846014.

Etymology: “multiappendiculata” refers to the conidia of this species containing multiple appendages.

Holotype: KUN-HKAS 122866.

*Saprotrophic* on submerged decaying wood in a freshwater lake. Sexual morph: undetermined. Asexual morph: hyphomycetous. *Colonies* on a natural substrate were punctiform, sporodochial, scattered and brown. *Mycelium* was composed of immersed or partly superficial, pale brown, septate, smooth, thin-walled and branched hyphae. *Conidiophores* were micronematous, mononematous, septate, cylindrical, pale brown, smooth and thin-walled. *Conidiogenous cells* were (10–)11–18(–20) × (–3)4–8(–10) ($\bar{x}$ = 14 × 6 μm, n = 20) μm holoblastic, cylindrical to subglobose, hyaline to pale brown, smooth, thin-walled and easy to break. *Conidia* were (50–)55–62(–68) × (–16)19–22(–24) μm ($\bar{x}$ = 59 × 20 μm, n = 50), solitary, cheiroid, cylindrical, pale brown, non-complanate, smooth, closely appressed, composed of (5–)7 rows of cells, euseptate, each row with 9–13 cells, slightly bent inward at the apex, straight or slightly curved arms were inserted in different planes, with 1–5(–6) hyaline, globose to subglobose subapical appendages, and appendages were thin and easy to rupture and shrink. Conidial secession was schizolytic.

Culture characteristics: Conidia germinated on PDA within 12 h and germ tubes were produced at the basal cell (Figure 4k). Colonies on PDA reached about 4 cm in 1 month at room temperature. Mycelium was loose, flocculent, smooth-edged, white to light yellow on the surface and pale brown to brown in reverse.

Material examined: China, Yunnan Province, Yuxi City; on submerged decaying wood in Fuxian lake; 24°30′33″ N, 102°54′36″ E, (1700 m); 10 July 2021; H.W. Shen and Q.X. Yang; and L-963 (KUN-HKAS 122866, holotype) and ex-type cultures (KUNCC 22-10734). Furthermore, *ibid*.; 24°37′14″ N 102°51′03″ E, (1700 m); 12 July 2021; Y.K. Jiang and S. Luan; and L-987 (KUN-HKAS 122870) and living cultures (KUNCC 22-10736).

Notes: *Dictyocheirospora multiappendiculata* shared similar morphological characters with other *Dictyocheirospora* species in having cheiroid, cylindrical, euseptate and non-complanate conidia that were densely clustered at the apex. The globose to subglobose apical or subapical appendages were similar to *D. hydei*, *D. indica*, *D. musae*, *D. nabanheensis*, *D. pseudomusae*, *D. suae* and *D. tetraploides*. However, *D. multiappendiculata* was distinguished from these species by the numbers of appendages (Table 2). In the phylogenetic analysis, *D. multiappendiculata* and *D. suae* formed a sister lineage with strong support (95% ML/1.00 PP, Figure 1). For the nucleotide comparison, see the Notes section of *D. suae*. Following Jeewon and Hyde’s [75] recommendations for establishing species boundaries and new taxa among fungi, we introduce *D. multiappendiculata* as a new species here.

*Dictyocheirospora aquadulcis* Sorvongxay, S. Boonmee & K.D Hyde, Fungal Diversity 96: 23 (2019), Figure 5.

Index Fungorum number: IF556308; Facesoffungi number: FoF05963.

Material examined: Thailand, Chiang Rai Province, Muang Chiang Rai District; 20°00′14″ N, 99°43′01″ E; on submerged decaying wood in a stream; 2 January 2021; H.W. Shen; and CR 8-29 (MFLU 22-0201) and living culture (MFLUCC 22-0095).

Notes: Phylogenetic analysis showed that our new collection MFLUCC 22-0095 clustered with the type strain of *Dictyocheirospora aquadulcis* (MFLUCC 17-2571) with low support (0.97% PP, Figure 1). Our taxon fit well with the morphological characteristics of *D. aquadulcis* in having cheiroid, ellipsoid to cylindrical and euseptate conidia consisting of seven rows of cells. However, our strain (MFLUCC 22-0095) comprised hyaline globose to subglobose appendages at the conidia apical or subapical, which are lacking in *D. aquadulcis* [35]. There were 3 bp nucleotide (491/494, including 2 bp of gaps) differences of ITS between *D. aquadulcis* (MFLUCC 22-0095) and *D. aquadulcis* (MFLUCC 17-2571). Based on the morphological and phylogenetic analysis, we therefore identified our new collected strain as *D. aquadulcis*.

*Dictyocheirospora nabanheensis* Tibpromma & K.D. Hyde, Fungal Diversity 93: 10 (2018), Figure 6.

Index Fungorum number: IF554474; Facesoffungi number: FoF04483.

Material examined: Thailand, Chiang Rai Province, Muang Chiang Rai District; 20°00′1″ N, 99°43′01″ E; on submerged decaying wood in a stream; 2 Jan 2021; H.W. Shen; and CR 8-23 (MFLU 22-0200) and living culture (MFLUCC 22-0094).

Notes: Based on the phylogenetic analysis of the combined SSU, ITS, LSU and *tef 1-α* sequence data, our new collection (MFLUCC 22-0094) clustered with the ex-type strain of *Dictyocheirospora nabanheensis* (KUMCC 16-0152) with high support (99% ML/1.00 PP, Figure 1). Morphologically, this new collection resembled the holotype of *D. nabanheensis* in having hyaline globose to subglobose appendages in the apical region [9] but with a greater number of appendages (1–5 vs. 1–2). We identified our collection as *D*. *nabanheensis*. *Dictyocheirospora nabanheensis* was found on dead leaves of *Pandanus* sp. in a terrestrial habitat in Yunnan, China, whereas our new collection was found on submerged decaying wood in a freshwater habitat in Thailand. Our study expanded the geographical distribution of this species and showed that it adapted to different ecological niches.

*Dictyocheirospora heptaspora* (Garov.) M.J. D’souza, Boonmee & K.D. Hyde, Fungal Diversity 80: 469 (2016), Figure 7.

Index Fungorum number: IF 551589.

Material examined: Thailand, Chiang Rai Province, Muang Chiang Rai District; 20°00′14″ N, 99°43′01″ E; on submerged decaying wood in a stream; 2 January 2021; H.W. Shen; and CR 8-30 (MFLU 22-0202) and living culture (MFLUCC 22-0096).

Notes: Multigene phylogenetic analysis showed that *Dictyocheirospora heptaspora* was related to *D. aquadulcis*. *Dictyocheirospora heptaspora* was transferred from *Dictyosporium* to *Dictyocheirospora* by Boonmee et al. [5] based on the phylogenetic evidence of ITS sequences data. Tsui et al. [24] provided ITS and SSU sequence data for strain CBS 395.59 but there is no morphological description support for this strain. Hongsanan et al. [76] provided the ITS, LSU and TEF sequence data for *D. heptaspora* (DLUCC 1992) collected from submerged decaying wood in Yunnan, China. Morphologically, *D. aquadulcis* is almost identical to *D. heptaspora.* Goh et al. [22], Prasher and Verma [26], and Hongsanan et al. [76] provided a detailed morphological description for *D. heptaspora*, where the conidia are characterized by being cheiroid, ellipsoid to cylindrical, consisting of mostly 7 (5–7) rows of cells closely appressed together, each row composed of 10–18 cells, 50–86 µm long and 19–30 µm wide. Based on the comparative analysis of the ITS gene region, the nucleotide difference between each strain of *D. heptaspora* and *D. aquadulcis* strains was less than 1.5%, and thus, they should be merged into one species following the recommendations of Jeewon and Hyde [75]. However, we consider that the description of the type strain of *D. heptaspora* is vague and there is no sequence data. Therefore, the holotype of *D. heptaspora* needs to be examined to determine whether they belong to the same species.

The morphological characteristics of our new collection fit well with *D. heptaspora* [5,22]. In addition, the phylogenetic analysis showed that a new collection MFLUCC 22-0096 clustered with isolates of *D. heptaspora* (CBS 396.59 and DLUCC 1992). Although *Dictyocheirospora heptaspora* was isolated several times from submerged wood in Thailand, none of the previous strains were sequenced [22]; we provide the sequences for this species.

*Dictyosporium* Corda, Beiträge zur gesammten Natur- und Heilwissenschaften: 87 (1836).

Notes: Members of the *Dictyosporium* are distributed worldwide and are mainly saprotrophic on dead wood, decaying leaves and plant litter from terrestrial and aquatic habitats [10,23,24,25,26,27]. *Dictyosporium* currently contains 54 species, but there are still no molecular sequence data for most species. In this study, we discovered *Dictyosporium tubulatum* for the first time in China on submerged decaying wood in a small stream in Yiliang County, Yunnan Province, and provided ITS, LSU, SSU and *tef 1-α* sequence data.

*Dictyosporium tubulatum* J. Yang, K.D. Hyde & Z.Y. Liu, MycoKeys 36: 94 (2018), Figure 8.

Index Fungorum number: IF450470; Facesoffungi number: FoF04677.

*Saprotrophic* on submerged decaying wood in a freshwater habitat. Sexual morph: undetermined. Asexual morph: hyphomycetous. *Colonies* on a natural substrate were punctiform, sporodochial, scattered and dark brown to black. *Mycelium* was composed of partly immersed, partly superficial, hyaline to pale brown, septate and branched hyphae. *Conidiophores* were micronematous, mononematous, septate, cylindrical, pale brown to brown, smooth and thin-walled. *Conidiogenous cells* were (3–)5–10(–11) × 2–5(–7) μm ($\bar{x}$ = 8 × 3 μm, *n* = 30), monoblastic, cylindrical, terminal, determinate, septate, pale brown to brown, smooth, thin-walled and sometimes swollen. *Conidia* were (18–)21–26(–27) × (7–)12–15(–18) μm ($\bar{x}$ = 23 × 14 μm, *n* = 50), acrogenous, solitary, brown to dark brown, cheiroid, complanate, composed of four arms close together, 8–11 euseptate in each arm, side arms lighter than the middle arm, with hyaline, tubular, elongated appendages, (12–)14–20(–27) × (3–)4–5(–6) μm ($\bar{x}$ = 17 × 4 μm, *n* = 40) and attached at the apical part of two outer arms. *Conidial secession* was schizolytic.

Material examined: China, Yunnan Province, Kunming City, Yiliang County; 24°39′27″ N, 103°08′05″ E, (2100 m); on submerged decaying wood in a small stream; 16 May 2021; H.W. Shen; YL 3-72-1 (KUN-HKAS 115789). We tried to obtain pure cultures on common fungal isolation media (PDA, MEA, WA, CMA), but its conidia did not germinate on any of the media. We therefore obtained DNA sequences directly from clean colonies on natural substrates.

Notes: Multigene phylogenetic analysis showed that our new collection *Dictyosporium tubulatum* (KUN-HKAS 115789) clustered with isolates of *D. tubulatum* (MFLUCC 15-0631 and MFLUCC 17-2056). Morphologically, our collection fit well with the description of a holotype (MFLU 15-1166) in having four-armed, cheiroid and complanate conidia with hyaline, tubular and elongated appendages [10]. Based on the morphological characteristics and phylogenetic support, we identified the new collection as *D. tubulatum*. The type strain of *Dictyosprium tubulatum* (MFLUCC 15-0631) was collected from a freshwater habitat in Thailand, while our new collection was collected from freshwater habitat in China. It is a new record for China; therefore, our study suggested that this species was distributed geographically.

*Digitodesmium* P.M. Kirk, Transactions of the British Mycological Society 77: 284 (1981).

Notes: *Digitodesmium* accommodated a group of species with divergent/closely gathered conidial rows, including *D. bambusicola*, whose conidia produced only three rows on the same plane [42]. Based on morphological features and phylogenetic analysis, we introduce another species here, viz., *Digitodesmium aquaticum*, that produced three rows in the same plane.

*Digitodesmium aquaticum* H.W. Shen, S. Boonmee & Z.L. Luo, sp. nov., Figure 9.

MycoBank number: MB 846307.

Etymology: “aquaticum” refers to the aquatic environment where this species was collected.

*Saprotrophic* on submerged decaying wood in a freshwater habitat. Sexual morph: undetermined. Asexual morph: hyphomycetous. *Colonies* on a natural substrate were punctiform, sporodochial, scattered, dark brown and white, and flaky. *Mycelium* were composed of partly immersed, partly superficial, hyaline to pale brown, septate and branched hyphae. *Conidiophores* were micronematous, mononematous, septate, cylindrical, hyaline to pale brown, thin-walled and unbranched. *Conidiogenous cells* were (3–)5–7(–9) × 3–4(–5) μm ($\bar{x}$ = 6 × 4 μm, *n* = 35), monoblastic, holoblastic, cylindrical sometimes flat at the base, septate, hyaline to pale brown, smooth and thin-walled. *Conidia* were (39–)41–44(–46) × (17–)19–21(–22) μm ($\bar{x}$ = 43 × 20 μm, *n* = 35), acrogenous, solitary, cheiroid, brown, complanate, composed of three arms, 6–9 euseptate in each arm, truncated at the basal cell, and slightly bent inward and hyaline at the apex cell of each arm. *Conidial secession* was schizolytic.

Material examined: Thailand, Chiang Rai Province, Muang Chiang Rai District; 19°57′55″ N, 99°41′10″ E; on submerged decaying wood in a stream; 16 January 2021; H.W. Shen; and CR 9-6 (MFLU 22-0203, holotype). We tried to obtain pure cultures on common fungal isolation media (PDA, MEA, WA, CMA) but its conidia did not germinate on any of the media. We therefore obtained DNA sequences directly from clean colonies on natural substrates.

Notes: The phylogenetic analysis showed that our collection *Digitodesmium aquaticum* (MFLU 22-0203) clustered with a type strain of *Di. bambusicola* (CBS 110279) and unidentified species *Di.* sp. (TBRC 10,037 and TBRC 10038) with strong support (100% ML and 1.00 PP, Figure 1) but formed a distinct lineage. Comparison of the ITS and LSU sequences data of *Di. aquaticum* and *Di. bambusicola* showed 99.39% (489/492 bp) and 99.24% (521/525 bp, including one gap) sequence identity, respectively. Morphologically, *Digitodesmium aquaticum* resembled *Di. bambusicola* in having cheiroid, euseptate and complanate conidia. However, *Di. bambusicola* (CBS 110279) had smaller conidia (24–32.5 × 12.5–23 μm vs. 41–44 × 19–21 μm), with hyaline, thin-walled, globose to subglobose or clavate appendages, and apical or subapical on the arms, which was distinct from our new species [42]. *Digitodesmium bambusicola* (PDD 74494) was collected from the submerged bamboo culms in a river in Philippines, while our new species was collected from submerged decaying wood in a stream in Thailand. Therefore, *Digitodesmium aquaticum* can be recognized as a phylogenetically distinct species and described in this study.

*Vikalpa* D’souza, Boonmee, Bhat & K.D. Hyde, Fungal Diversity 80: 479 (2016).

Notes: *Vikalpa* has four species, and currently only the type species *Vikalpa australiensis* HKUCC 8797 has unique ITS sequence data. This study contributed two additional novel species from freshwater habitats, viz., *V. grandispora* and *V. sphaerica*. Detailed descriptions and colored photoplates are given below.

*Vikalpa grandispora* H.W. Shen, S. Boonmee & Z.L. Luo, sp. nov., Figure 10.

MycoBank number: MB 846015.

Etymology: “*grandispora*” refers to the large conidia.

Holotype: KUN-HKAS 122867.

*Saprotrophic* on submerged decaying wood in a freshwater habitat. Sexual morph: undetermined. Asexual morph: hyphomycetous. *Colonies* on a natural substrate were punctiform, sporodochial, scattered and brown. *Mycelium* were composed of immersed, septate, branched and hyaline to pale brown hyphae. *Conidiophores* were micronematous, mononematous, septate, subglobose to cylindrical, unbranched, hyaline to pale brown, smooth, thin-walled and sometimes reduced to conidiogenous cells. *Conidiogenous cells* were holoblastic, subglobose, smooth and thin-walled. *Conidia* were (33–)38–49(–53) × (9–)13–18(–21) μm ($\bar{x}$ = 44 × 16 μm, *n* = 50), solitary, cheiroid, pale brown to brown, not complanate, with 3 rows in different planes, arms closely appressed when young, separated at maturity, 9–12 cells in each row (5–)6 × 7 μm wide, euseptate, irregular, constricted at the septa and guttulate; appendages were more inflated than the apical cell, (4–)5–6(–7) × (5–)6–7(–9) μm ($\bar{x}$ = 5 × 7 μm, *n* = 50), hyaline and globose to subglobose. *Conidial secession* was schizolytic.

Culture characteristics: Conidia germinated on PDA within 12 h and germ tubes were produced at the basal cell (Figure 10j). Colonies grew on PDA; mycelium grew slowly, reaching about 2.5 cm in 1 month at room temperature and were loose, flocculent, smooth, white to creamy-yellow on the surface, orange on the edge and gray in the middle on the reverse side.

Material examined: China, Yunnan Province, Lijiang City, Yadong village; 26°58′08″ N, 100°24′29″ E, (2160 m); on submerged decaying wood in a small stream; 20 October 2021; H.W. Shen; and H593 (KUN-HKAS 122867, holotype) and ex-type culture (KUNCC 22-12425).

Notes: *Vikalpa grandispora* resembled *V. australiensis*, *V. freycinetiae*, *V. micronesiaca* and *V. sphaerica* in having solitary, cheiroid, non-complanate and three-armed conidia [46,47,48]. *Vikalpa grandispora* could be easily distinguished from other species by its larger conidia (Table 2). The phylogenetic analysis of the combined SSU, ITS, LSU and *tef1-α* sequence data showed that *V. grandispora* clustered as a sister lineage to *V. sphaerica* with low support (84% ML, Figure 1). The nucleotide comparison of the SSU, ITS, LSU and *tef1-α* genes of *V. grandispora* and *V. sphaerica* revealed 9 bp (2%), 5 bp (0.6%, including gaps), 5 bp (0.9%, including gaps) and 18 bp (2%) nucleotide differences, respectively. Therefore, we identified *V. grandispora* as a new species in *Vikalpa* based on the distinguished morphology and phylogenetic evidence.

*Vikalpa sphaerica* H.W. Shen & Z.L. Luo, sp. nov., Figure 11.

MycoBank number: MB 846016.

Etymology: “sphaerica” refers to the spherical appendages.

Holotype: KUN-HKAS 115805.

*Saprotrophic* on submerged decaying wood in a freshwater habitat. Sexual morph: undetermined. Asexual morph: hyphomycetous. *Colonies* on a natural substrate were punctiform, sporodochial, scattered and dark brown. *Mycelium* were composed of immersed or partly superficial, septate, branched and hyaline to pale brown hyphae. *Conidiophores* were micronematous, mononematous, septate, cylindrical, unbranched, hyaline to pale brown, smooth-walled and sometimes reduced to conidiogenous cells. *Conidiogenous cells* were 4–7(–9) × (2–)3–4(–5) (M = 5 × 4 μm, *n* = 15), holoblastic, cylindrical, sometimes flat at the base, septate, hyaline to pale brown and smooth-walled. *Conidia* were (23–)26–30(–34) × (11–)16–19(–20) μm ($\bar{x}$ = 28 × 17 μm, *n* = 50), solitary, cheiroid, pale brown to brown, not complanate, usually composed of 3–4 rows, rarely with 3 rows, rows separated from each other, 5–9 cells in each row, euseptate, irregular, constricted at the septa, guttulate, with hyaline, globose to subglobose appendages (6–)7–8(–9) × (5–)7–8(–9) μm ($\bar{x}$ = 8 × 8 μm, *n* = 50) at the apical cells and a few rows had either no appendages or they had fallen off. *Conidial secession* schizolytic.

Culture characteristics: Conidia germinated on PDA within 12 h and germ tubes were produced at both ends (Figure 11m). Colony growth on PDA; growth was slow, reaching about 4 cm in 1 month at room temperature. Mycelium was loose, flocculent, smooth edged, brownish-red on the forward, orange on the edge and gray in the middle on the reverse side.

Material examined: China, Yunnan Province, Lijiang City, Luguhu Lake; 27°42′11″ N, 100°48′18″ E, (2700 m); on submerged decaying wood in a small stream; 04 March 2021; Z.Q. Zhang and L. Sha; and L185 (KUN-HKAS 115805, holotype) and ex-type cultures (CGMCC3.20682 = KUNCC 21-10711).

Notes: Phylogenetic analysis placed our new taxon *Vikalpa sphaerica* (CGMCC 3.20682) close to *V. grandispora* with low support (84% ML, Figure 1). Morphologically, *V. sphaerica* resembled *V. grandispora*, *V. australiensis* and *V. freycinetiae* in having solitary and cheiroid conidia with apical appendages [48]. However, *V. grandispora* had longer conidia (38–49 vs. 26–30) and more cells in each row (9–12 vs. 5–9). The rows of *V sphaerica* were separated from each other and had fewer cells in each row than *V. australiensis* (5–9 vs. 7–11), while *V. freycinetiae* had a greater number of cells (9–13 vs. 5–9) that were larger (31–43 μm vs. 26–30 μm) than *V. sphaerica* [46]. Therefore, *V. sphaerica* is introduced here as a new species based on the morphological and phylogenetic analysis.

## 4. Discussion

*Dictyosporiaceae* accommodates a holomorphic group of *Dothideomycetes* that can produce cheiroid (digitate) and septate conidia [5]. *Dictyosporaceae* was recently well-studied based on the combination of morphological and phylogenetic analysis. However, due to the lack of valuable cultures and molecular sequence data, the taxonomic status of several earlier-described genera with fewer members remains confusing and ambiguous. Therefore, morphological identification continues to play a crucial role in the identification of species in *Dictyosporiaceae*.

DNA sequence data are available for most cheirosporous genera, except *Dictyopalmispora*. Therefore, the distinction between *Dictyopalmispora* and other cheirosporous genera is mainly based on unique hair-like appendages produced on all arms of *Dictyopalmispora* [5]. Four species are accepted in *Vikalpa*, including *V. lignicola* from freshwater environments, and only *V. australiensis* has ITS sequence data. The species of *Vikalpa* are distinguished from other genera mainly by the three rows of conidia on different planes produced in sporododochial conidiomata [5]. In this study, *V. grandispora* and *V. sphaerica* were introduced from freshwater lotic and lentic environments to enrich the habitat and distribution area of *Vikalpa* species.

The species of *Digitodesmium* is morphologically confused with *Dictyosporium*, but some morphological features can be used to distinguish them from each other: the arms of *Digitodesmium* are separated at the apex, while those in *Dictyosporium* are not separated. In addition, the conidial secession of *Digitodesmium* is schizolytic, while it is rhexolytic in *Dictyosporium* [6,42,43]. *Digitodesmium polybrachiatum* is easily confused with the *Dictyocheirospora* species in having non-complanate, cheiroid conidia and the arms being closely compacted at the apex. Therefore, using morphological characteristics alone, they cannot be properly distinguished [44]. Multigene phylogeny showed that members of *Digitodesmium* are divided into two main clades: Clade 1 is composed of *Di. bambusicola* and *Di. aquaticum*, with the main feature being that the conidia are produced in sporodochia, consisting of three rows of cells on one plane and they are euseptate. Clade 2 is composed of *Di. chiangmaiense* and *Di. polybrachiatum,* which are mainly characterized by their conidia arms being closely gathered at the apex, and they are euseptate, with or without appendages. There is a clear phylogenetic distance between these two clades, which strongly suggests that *Di. bambusicola* and *Di. aquaticum* belong to a different genus. However, there are no valuable culture and sequence data for the five species in the genus, including type species *Di. elegans*, to ensure which clade has been adapted as the type species; therefore, we agree with the suggestions of Nobrega et al. [44] and still maintain the taxonomic status of the species in the two clades.

Cheirosporous hyphomycetes are widely distributed as saprotrophic fungi on various plant debris substrates in freshwater and terrestrial habitats worldwide [5,6,7,8,9,10,11,12,13,14]. Although cheirosporous hyphomycetes are distributed globally, the main species contribution is from the Greater Mekong Subregion (GMC, Thailand and Yunnan, China) in Asia, accounting for more than 50% of the total. The Greater Mekong Subregion is rich in cheirosporous hyphomycetes resources, and species on wood substrates in freshwater habitats account for more than 50% of the total in this region. We are conducting research on lignicolous freshwater fungi in this region, which provides insights into the lignicolous freshwater taxa and increases knowledge of microfungi in the Greater Mekong Subregion.

## Figures and Tables

**Figure 1 jof-08-01200-f001:**
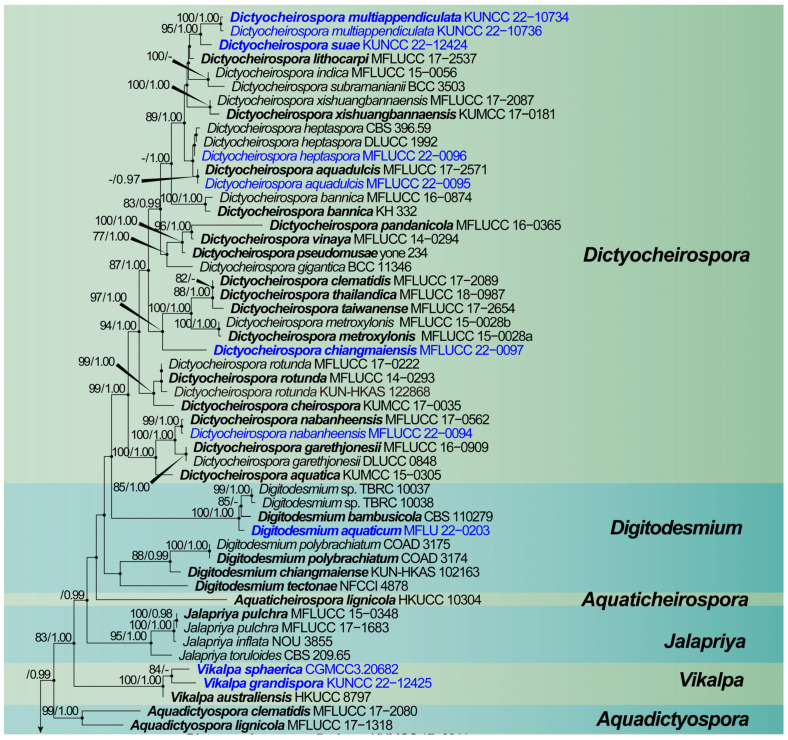
The maximum likelihood (ML) tree is based on the combined SSU, ITS, LSU and *tef 1-α* sequence data. Bootstrap support values with an ML greater than 75% and Bayesian posterior probabilities (PP) greater than 0.97 are given above the nodes, shown as “ML/PP”. The tree was rooted to *Periconia igniaria* (CBS 379.86 and CBS 845.96). New species are indicated in blue and type strains are in bold.

**Figure 2 jof-08-01200-f002:**
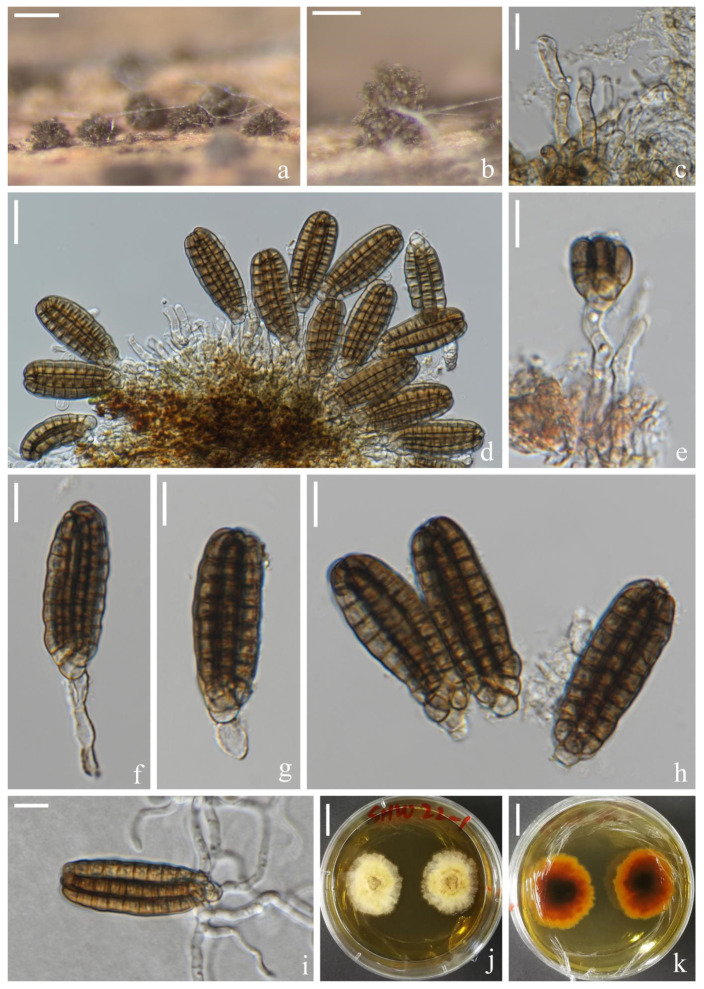
*Dictyocheirospora chiangmaiensis* (MFLU 22–0199, holotype). (**a**,**b**) Colonies on the substratum. (**c**) Conidiophores. (**d**) Conidiomata with conidiophores. (**e**–**g**) Conidiogenous cells with conidia. (**h**) Conidia. (**i**) Germinating conidium. (**j**,**k**) Culture colonies on MEA, reverse (left) and obverse (right). Scale bars: (**a**) 200 μm, (**b**) 100 μm, (**c**,**e**–**i**) 10 μm, (**d**) 20 μm and (**j**,**k**) 1 cm.

**Figure 3 jof-08-01200-f003:**
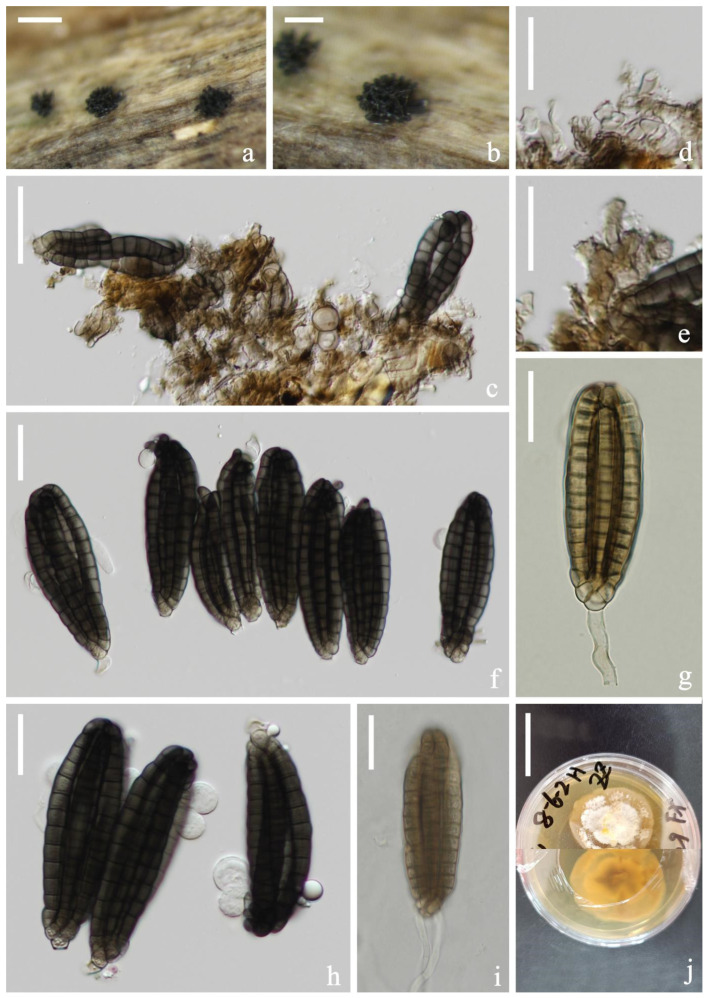
*Dictyocheirospora suae* (KUN-HKAS 121703, holotype). (**a**,**b**) Colonies on the substratum. (**c**) Conidiomata. (**d**,**e**) Conidiophores and conidiogenous cells. (**f**) Conidia. (**g**) Conidiogenous cells with conidia. (**h**) Conidia with appendages. (**i**) Germinating conidium. (**j**) Culture colonies on PDA, reverse (upper) and obverse (lower). Scale bars: (**a**) 200 μm, (**b**) 100 μm, (**c**) 30 μm, (**d**–**i**) 20 μm and (**j**) 2 cm.

**Figure 4 jof-08-01200-f004:**
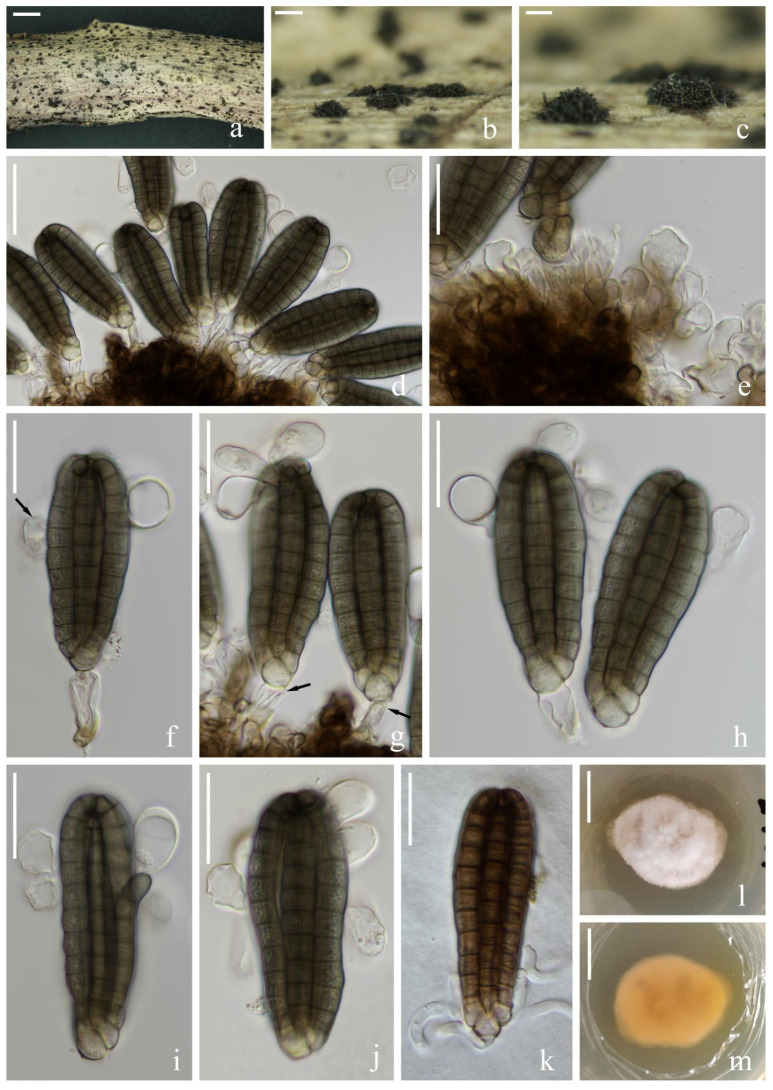
*Dictyocheirospora multiappendiculata* (KUN-HKAS 122866, holotype). (**a**–**c**) Colonies on the substratum. (**d**) Conidiomata and conidia. (**e**) Conidiophores and conidiogenous cells. (**f**–**h**) Conidiogenous cells, conidia and appendages. (**i**,**j**) Conidia with appendages. (**k**) Germinating conidium. (**l**,**m**) Culture colonies on PDA, reverse (upper) and obverse (lower). Scale bars: (**a**) 2000 μm, (**b**) 200 μm, (**c**) 100 μm, (**d**) 30 μm, (**e**–**k**) 20 μm and (**l**,**m**) 1 cm.

**Figure 5 jof-08-01200-f005:**
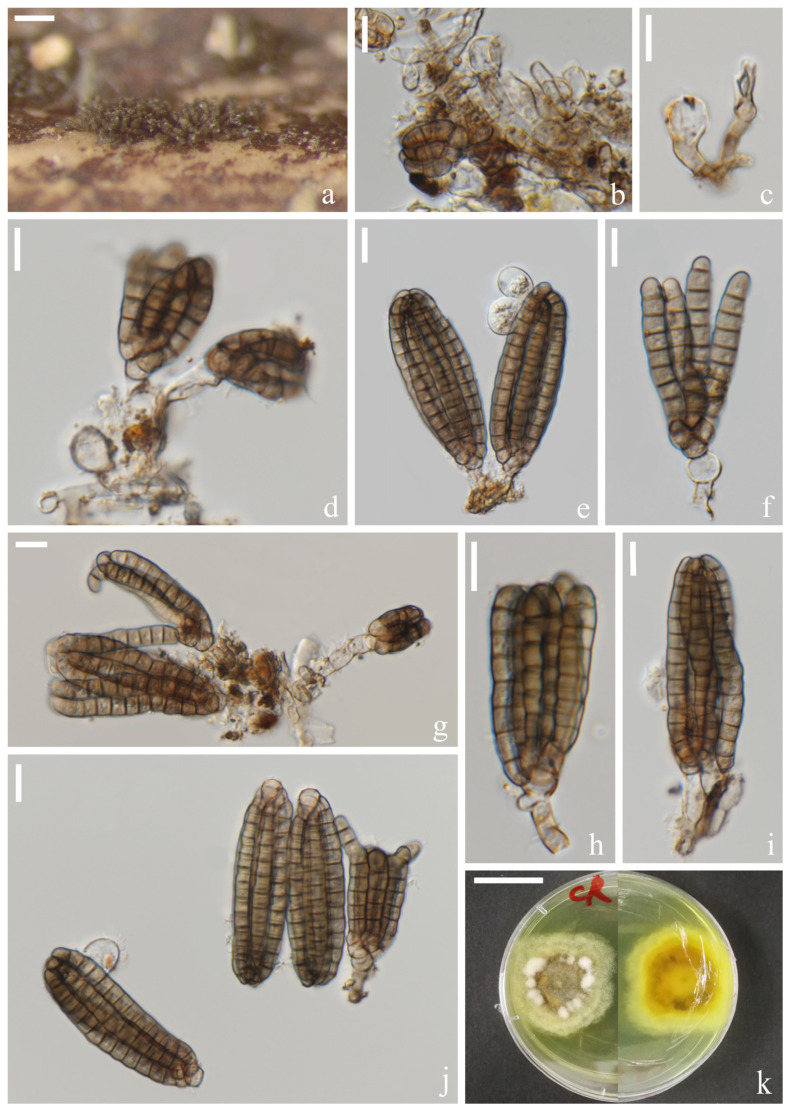
*Dictyocheirospora aquadulcis* (MFLU 22-0201). (**a**) Colonies on the substratum. (**b**) Conidiomata. (**c**) Conidiophores. (**d**,**e**) Conidiophores and conidia. (**f**) Conidiophores, conidia and appendages. (**g**–**i**) Conidiogenous cells and conidia. (**j**) Conidia. (**k**) Culture colonies on PDA, reverse (left) and obverse (right). Scale bars: (**a**) 100 μm, (**b**–**j**) 20 μm and (**k**) 2 cm.

**Figure 6 jof-08-01200-f006:**
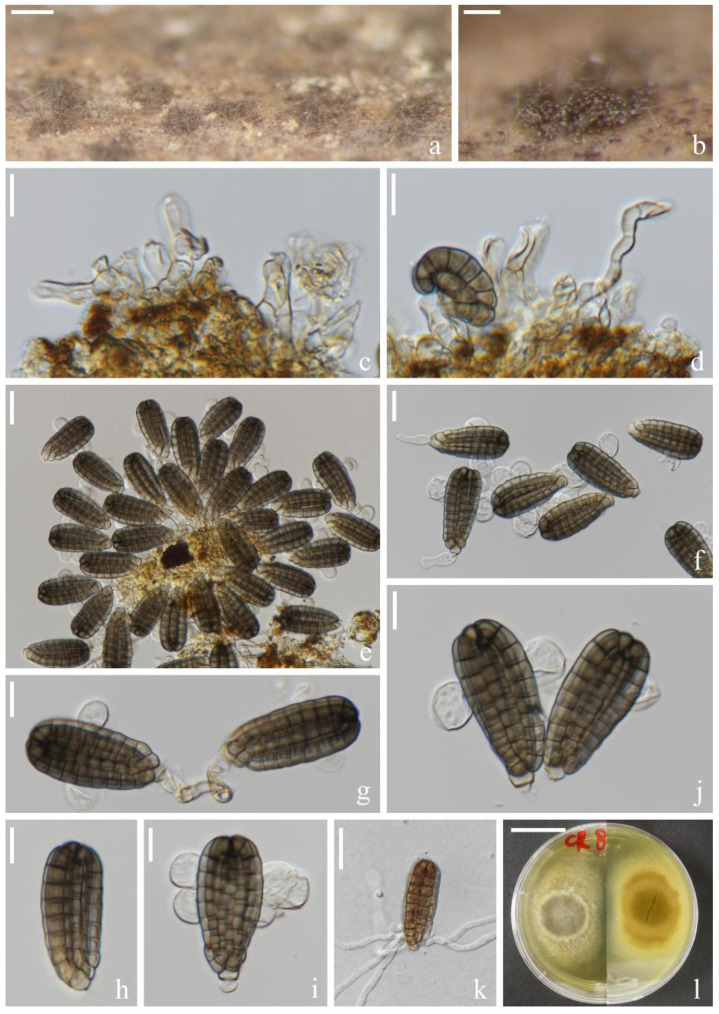
*Dictyocheirospora nabanheensis* (MFLU 22-0200). (**a**,**b**) Colonies on the substratum. (**c**,**d**) Conidiophores. (**e**) Conidia heap. (**f**,**g**) Conidiogenous cells, conidia and appendages. (**h**) Conidia. (**i**,**j**) Conidia with appendages. (**k**) Germinating conidium. (**l**) Culture colonies on PDA, reverse (left) and obverse (right). Scale bars: (**a**) 100 μm, (**b**) 200 μm, (**c**,**d**,**g**–**j**) 10 μm, (**e**,**f**,**k**) 20 μm and (**l**) 2 cm.

**Figure 7 jof-08-01200-f007:**
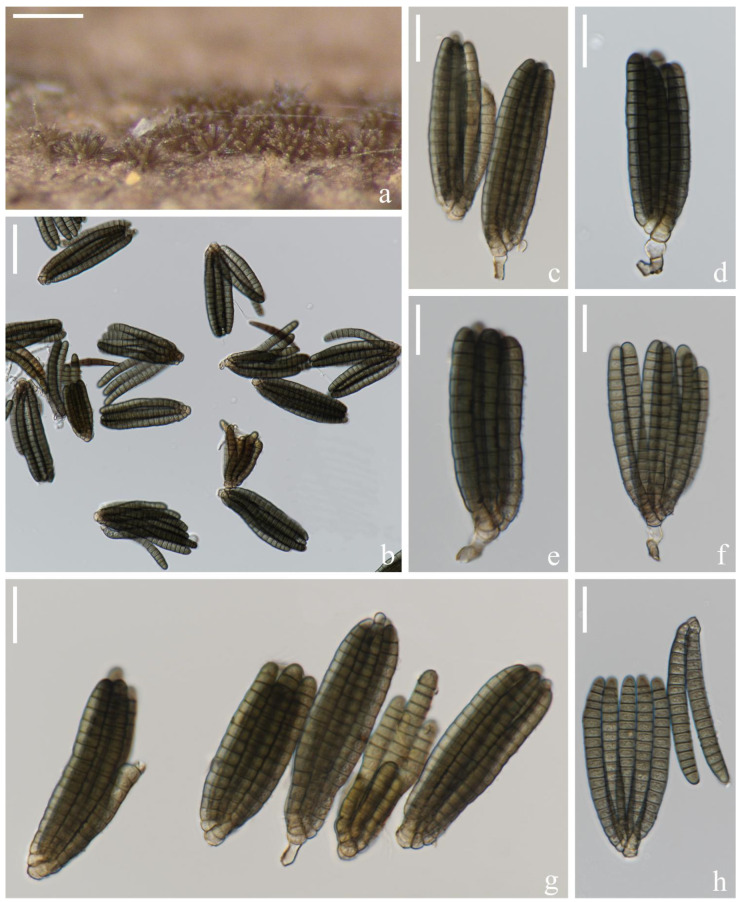
*Dictyocheirospora heptaspora* (MFLU 22-0202). (**a**) Colonies on the substratum. (**c**–**f**) Conidiogenous cells with conidia. (**b**,**g**,**h**) Conidia. Scale bars: (**a**) 200 μm, (**b**) 40 μm and (**c**–**h**) 20 μm.

**Figure 8 jof-08-01200-f008:**
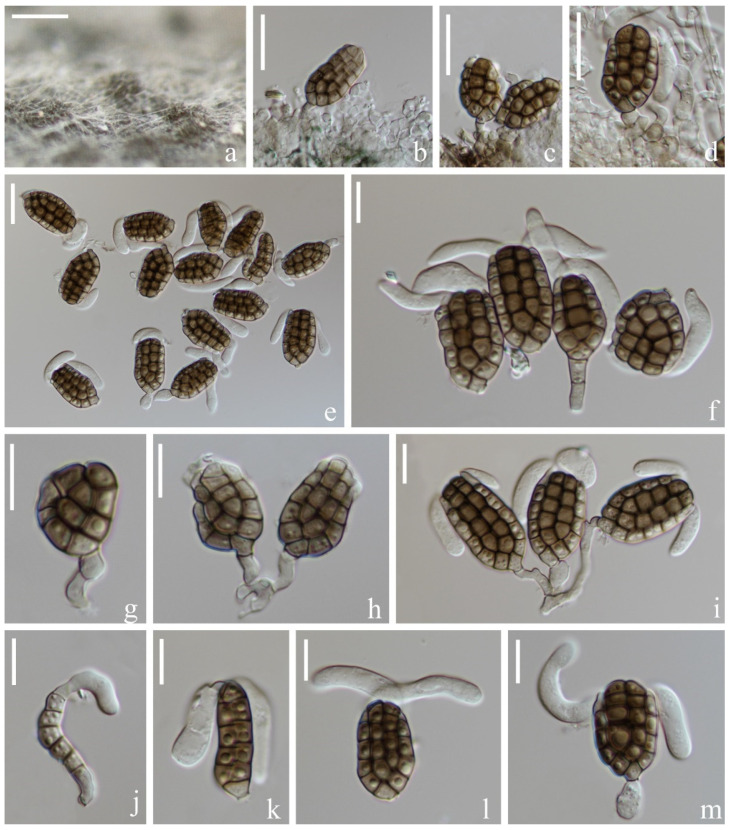
*Dictyosprium tubulatum* (KUN-HKAS 115789, new geographical record). (**a**) Colonies on the substratum. (**b**–**d**) Conidiophores and conidia. (**e**) Conidia heap. (**f**,**i**,**m**) Conidiogenous cells, conidia and appendages. (**g**,**h**) Conidiogenous cells with conidia. (**j**–**l**) Conidia with appendages. Scale bars: (**a**) 200 μm, (**b**–**e**) 20 μm and (**f**–**m**) 10 μm.

**Figure 9 jof-08-01200-f009:**
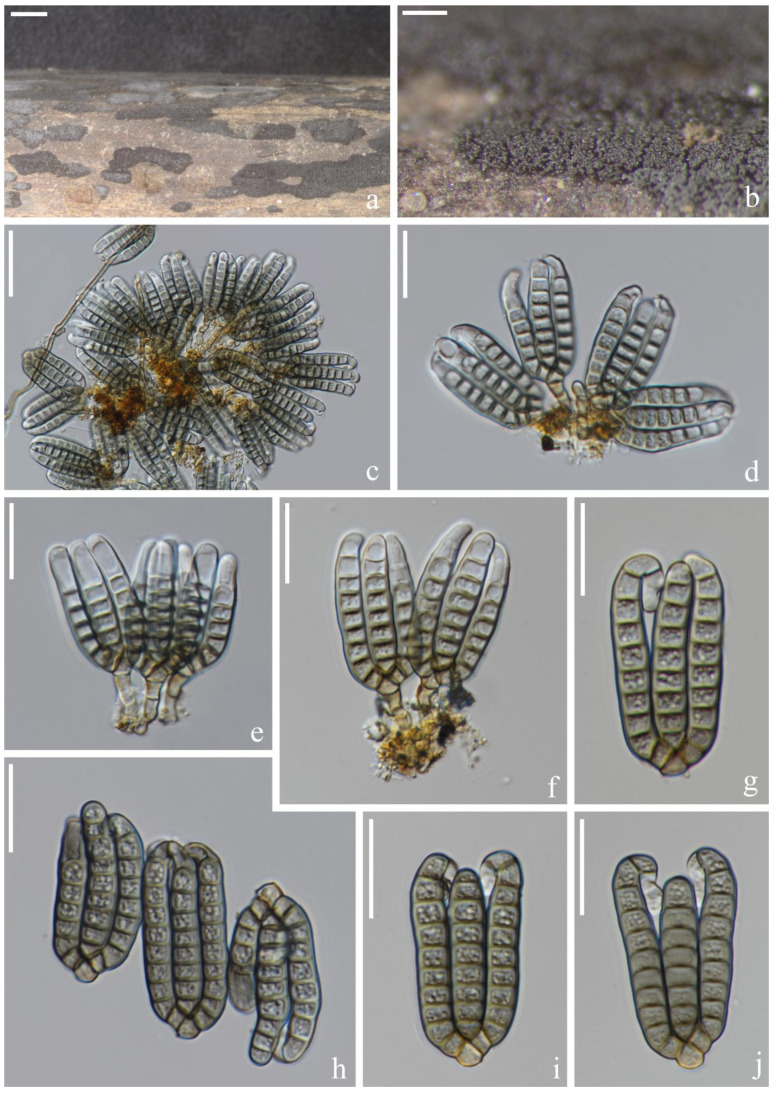
*Digitodesmium aquaticum* (MFLU 22-0203, holotype). (**a**,**b**) Colonies on the substratum. (**c**–**f**) Conidiophores with conidia. (**g**–**j**) Conidia. Scale bars: (**a**) 1000 μm, (**b**) 100 μm, (**c**) 40 μm and (**d**–**j**) 20 μm.

**Figure 10 jof-08-01200-f010:**
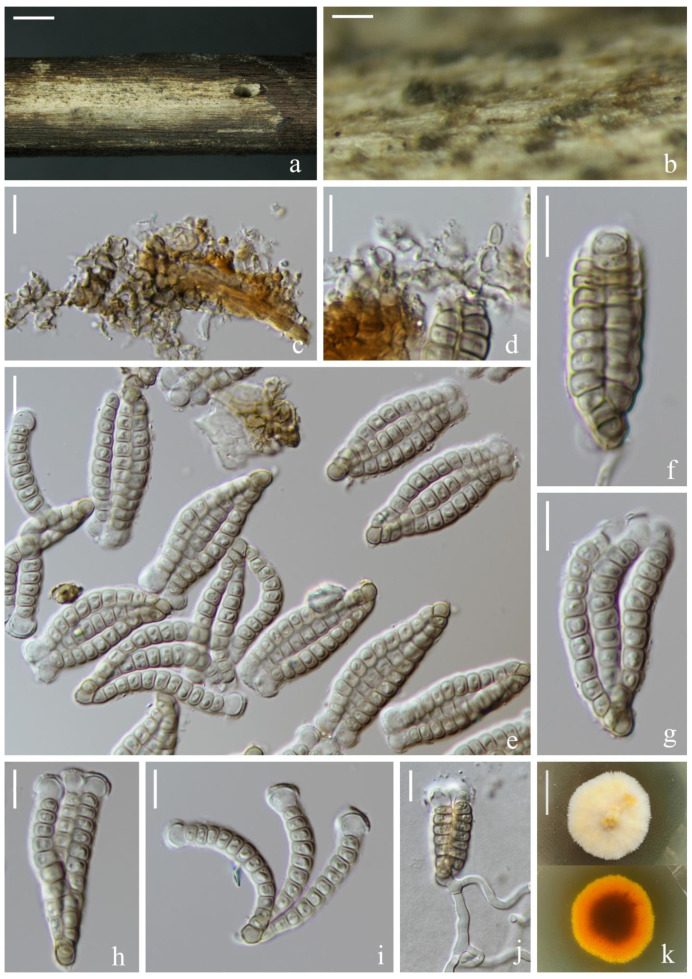
*Vikalpa grandispora* (KUN-HKAS 122867, holotype). (**a**,**b**) Colonies on the substratum. (**c**) Conidiomata. (**d**) Conidiogenous cells. (**e**,**g**–**i**) Conidia with appendages. (**f**) Conidia. (**j**) Germinating conidium. (**k**) Culture colonies on PDA, reverse (upper) and obverse (lower). Scale bars: (**a**) 1000 μm, (**b**) 200 μm, (**c**–**j**) 10 μm and (**k**) 1 cm.

**Figure 11 jof-08-01200-f011:**
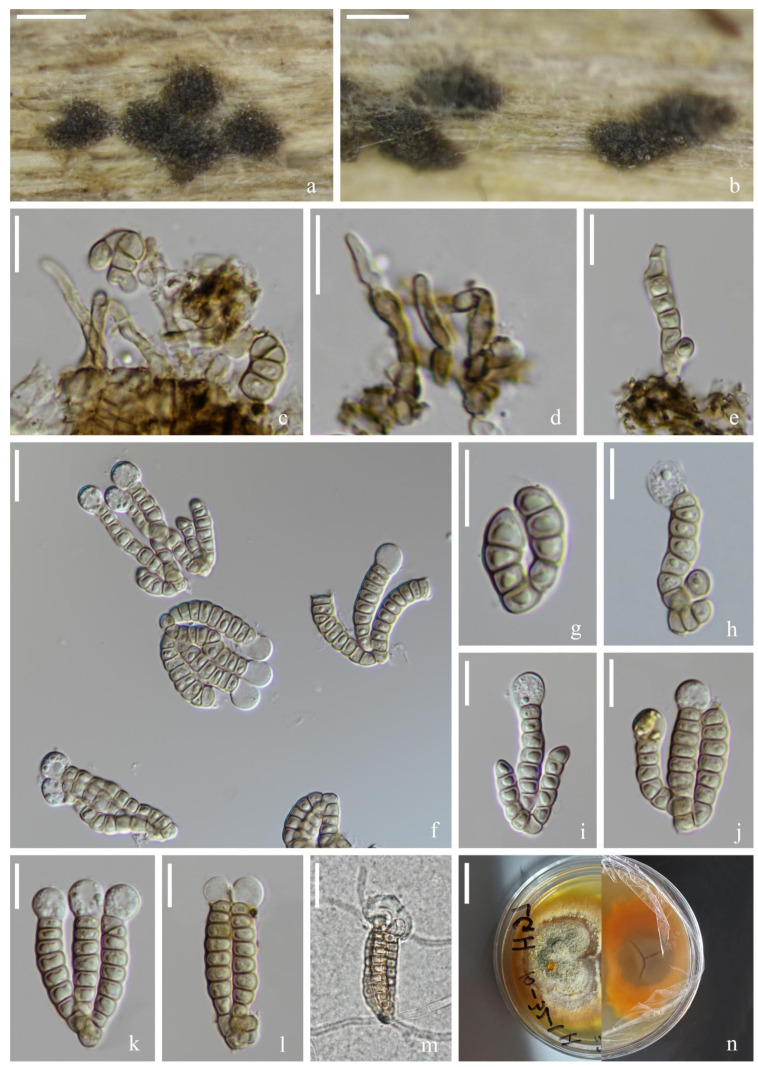
*Vikalpa sphaerica* (KUN-HKAS 115805, holotype). (**a**,**b**) Colonies on the substratum. (**c**,**d**) Conidiophores and conidiogenous cells. (**e**) Conidia. (**f**–**l**) Conidia with appendages. (**m**) Germinating conidium. (**n**) Culture colonies on PDA, reverse (left) and obverse (right). Scale bars: (**a**,**b**) 200 μm, (**c**–**m**) 10 μm and (**n**) 1 cm.

**Table 1 jof-08-01200-t001:** Strains/specimens used for phylogenetic analysis and their GenBank accession numbers.

Species ^1^	Source ^2^	GenBank Accession Number ^3^			
		SSU	ITS	LSU	*tef 1-α*
*Aquadictyospora clematidis*	MFLUCC 17-2080 ^T^	MT226664	MT310592	MT214545	MT394727
*Aquadictyospora lignicola*	MFLUCC 17-1318 ^T^	–	MF948621	MF948629	MF953164
*Aquaticheirospora lignicola*	HKUCC 10304 ^T^	AY736377	AY864770	AY736378	–
*Cheirosporium triseriale*	HMAS 180703 ^T^	–	EU413953	EU413954	–
*Dendryphiella phitsanulokensis*	MFLUCC 17-2513 ^T^	MG754402	MG754400	MG754401	–
*Dendryphiella variabilis*	CBS 584.96 ^T^	–	LT963453	LT963454	–
*Dendryphiella eucalyptorum*	CBS 137987 ^T^	–	KJ869139	KJ869196	–
*Dendryphiella fasciculata*	MFLUCC 17-1074 ^T^	–	MF399213	MF399214	–
*Dendryphiella paravinosa*	CBS 141286 ^T^	–	KX228257	KX228309	–
*Dictyocheirospora aquadulcis*	MFLUCC 17-2571 ^T^	–	MK634545	MK634542	–
*Dictyocheirospora aquadulcis* *	MFLUCC 22-0095	OP526625	OP526634	OP526644	OP542236
*Dictyocheirospora aquatica*	KUMCC 15-0305 ^T^	–	KY320508	KY320513	–
*Dictyocheirospora bannica*	KH 332 ^T^	AB797223	LC014543	AB807513	AB808489
*Dictyocheirospora bannica*	MFLUCC 16-0874	–	MH381765	MH381774	–
*Dictyocheirospora cheirospora*	KUMCC 17-0035 ^T^	MF928073	MF177035	MF177036	–
*Dictyocheirospora clematidis*	MFLUCC 17-2089 ^T^	MT226665	MT310593	MT214546	MT394728
*Dictyocheirospora chiangmaiensis* *	MFLUCC 22-0097 ^T^	OP526621	OP526630	OP526640	OP542232
*Dictyocheirospora garethjonesii*	MFLUCC 16-0909 ^T^	–	KY320509	KY320514	–
*Dictyocheirospora garethjonesii*	DLUCC 0848	–	MF948623	MF948631	MF953166
*Dictyocheirospora gigantica*	BCC 11346	–	DQ018095	–	–
*Dictyocheirospora heptaspora*	CBS 396.59	DQ018082	DQ018090	–	–
*Dictyocheirospora heptaspora*	DLUCC 1992	–	MT756244	MT756243	MT776563
*Dictyocheirospora heptaspora* *	MFLUCC 22-0096	–	OP526635	OP526645	OP542237
*Dictyocheirospora indica*	MFLUCC 15-0056	MH381757	MH381763	MH381772	MH388817
*Dictyocheirospora lithocarpi*	MFLUCC 17-2537 ^T^	MK347888	MK347781	MK347999	–
*Dictyocheirospora metroxylonis*	MFLUCC 15-0028a ^T^	MH742317	MH742321	MH742313	–
*Dictyocheirospora metroxylonis*	MFLUCC 15-0028b ^T^	MH742318	MH742322	MH742314	MH764301
*Dictyocheirospora multiappendiculata* *	KUNCC 22-10734 ^T^	OP526623	OP526632	OP526642	OP542234
*Dictyocheirospora multiappendiculata* *	KUNCC 22-10736	OP526624	OP526633	OP526643	OP542235
*Dictyocheirospora nabanheensis*	MFLUCC 17-0562 ^T^	–	MH388340	MH376712	MH388375
*Dictyocheirospora nabanheensis* *	MFLUCC 22-0094	OP526627	OP526637	OP526647	OP542239
*Dictyocheirospora pandanicola*	MFLUCC 16-0365 ^T^	MH388309	MH388341	MH376713	MH388376
*Dictyocheirospora pseudomusae*	yone 234 ^T^	AB797230	LC014550	AB807520	AB808496
*Dictyocheirospora rotunda*	MFLUCC 14-0293 ^T^	KU179101	KU179099	KU179100	–
*Dictyocheirospora rotunda*	MFLUCC 17-0222	–	MH381764	MH381773	MH388818
*Dictyocheirospora suae* *	KUNCC 22-12424 ^T^	OP526622	OP526631	OP526641	OP542233
*Dictyocheirospora subramanianii*	BCC 3503	–	DQ018094	–	–
*Dictyocheirospora taiwanense*	MFLUCC 17-2654 ^T^	–	MK495821	MK495820	–
*Dictyocheirospora thailandica*	MFLUCC 18-0987 ^T^	–	MT627734	MN913743	–
*Dictyocheirospora vinaya*	MFLUCC 14-0294 ^T^	KU179104	KU179102	KU179103	–
*Dictyocheirospora xishuangbannaensis*	KUMCC 17-0181 ^T^	MH388310	MH388342	MH376714	MH388377
*Dictyocheirospora xishuangbannaensis*	MFLUCC 17-2087	MT226666	MT310594	MT214547	MT394729
*Dictyosporium appendiculatum*	KUMCC 17-0311 ^T^	–	MH388343	MH376715	–
*Dictyosporium hongkongensis*	KMUCC 17-0268 ^T^	MH388313	MH388346	MH376718	MH388380
*Dictyosporium krabiense*	MFLU 16-1890 ^T^	MH388314	–	MH376719	MH388381
*Dictyosporium marinum*	GJ357 ^T^	–	–	MN017841	–
*Dictyosporium muriformis*	GZCC 20-0006 ^T^	MN901117	MT002304	MN897834	MT023011
*Dictyosporium pandanicola*	MFLU 16-1886 ^T^	–	MH388347	MH376720	MH388382
*Dictyosporium alatum*	ATCC 34953 ^T^	DQ018080	DQ018088	DQ018101	–
*Dictyosporium aquaticum*	MF 1318 ^T^	–	KM610236	–	–
*Dictyosporium bulbosum*	yone 221	AB797221	LC014544	AB807511	AB808487
*Dictyosporium digitatum*	KH 401	AB797225	LC014545	AB807515	AB808491
*Dictyosporium digitatum*	yone 280	AB797228	LC014547	AB807512	AB808488
*Dictyosporium elegans*	NBRC 32502 ^T^	DQ018079	DQ018087	DQ018100	–
*Dictyosporium guttulatum*	MFLUCC 16-0258 ^T^	MH388312	MH388345	MH376717	MH388379
*Dictyosporium hughesii*	KT 1847	AB797227	LC014548	AB807517	AB808493
*Dictyosporium meiosporum*	MFLUCC 10-0131 ^T^	KP710946	KP710944	KP710945	–
*Dictyosporium nigroapice*	BCC 3555	–	DQ018085	–	–
*Dictyosporium nigroapice*	MFLUCC 17-2053 ^T^	–	MH381768	MH381777	MH388821
*Dictyosporium olivaceosporum*	KH 375 ^T^	AB797224	LC014542	AB807514	AB808490
*Dictyosporium sexualis*	MFLUCC 10-0127 ^T^	KU179107	KU179105	KU179106	–
*Dictyosporium* sp.	MFLUCC 15-0629	–	MH381766	MH381775	MH388819
*Dictyosporium stellatum*	CCFC 241241 ^T^	–	NR_154608	JF951177	–
*Dictyosporium strelitziae*	CBS 123359 ^T^	–	NR_156216	FJ839653	–
*Dictyosporium tetrasporum*	KT 2865	AB797229	LC014551	AB807519	AB808495
*Dictyosporium thailandicum*	MFLUCC 13-0773 ^T^	–	KP716706	KP716707	–
*Dictyosporium tratense*	MFLUCC 17-2052 ^T^	MH381761	MH381767	MH381776	MH388820
*Dictyosporium tubulatum*	MFLUCC 15-0631 ^T^	–	MH381769	MH381778	MH388822
*Dictyosporium tubulatum*	MFLUCC 17-2056	–	MH381770	MH381779	–
*Dictyosporium tubulatum* *	KUN-HKAS 115789	OP749878	OP749871	OP749876	OP756063
*Dictyosporium wuyiense*	CGMCC 3.18703 ^T^	–	KY072977	–	–
*Dictyosporium zhejiangense*	MW-2009a ^T^	–	FJ456893	–	–
*Digitodesmium aquaticum* *	MFLU 22-0203 ^T^	OP749879	OP749872	OP749877	OP756064
*Digitodesmium bambusicola*	CBS 110279 ^T^	–	DQ018091	DQ018103	–
*Digitodesmium chiangmaiense*	KUN-HKAS 102163 ^T^	MK571775	–	MK571766	–
*Digitodesmium polybrachiatum*	COAD 3174 ^T^	MW879325	MW879318	MW879316	–
*Digitodesmium polybrachiatum*	COAD 3175	MW879326	MW879319	MW879317	–
*Digitodesmium* sp.	TBRC 10037	–	MK405234	MK405232	MK405230
*Digitodesmium* sp.	TBRC 10038	–	MK405235	MK405233	MK405231
*Gregarithecium curvisporum*	KT 922	AB797257	AB809644	AB807547	–
*Gregarithecium curvisporum*	MFLUCC 13-0853 ^T^	KX364283	KX364281	KX364282	–
*Immotthia atrograna*	ZT-Myc-64283	–	MW489540	–	–
*Immotthia bambusae*	KUN-HKAS 112012A1 ^T^	MW489461	MW489455	MW489450	MW504646
*Immotthia bambusae*	KUN-HKAS 112012B ^T^	–	MW489457	MW489452	–
*Immotthia bambusae*	KUN-HKAS 112012C ^T^	MW489463	MW489458	MW489453	MW504648
*Immotthia bambusae*	KUN-HKAS 112012D ^T^	MW489464	MW489459	MW489454	MW504649
*Jalapriya inflata*	NOU 3855	JQ267361	JQ267362	JQ267363	–
*Jalapriya pulchra*	MFLUCC 15-0348 ^T^	KU179110	KU179108	KU179109	–
*Jalapriya pulchra*	MFLUCC 17-1683	–	MF948628	MF948636	MF953171
*Jalapriya toruloides*	CBS 209.65	DQ018081	DQ018093	DQ018104	–
*Neodendryphiella mali*	FMR 16561 ^T^	–	LT906655	LT906657	–
*Neodendryphiella mali*	FMR 17003	–	LT993734	LT993735	–
*Neodendryphiella michoacanensis*	FMR 16098 ^T^	–	LT906660	LT906658	–
*Neodendryphiella tarraconensis*	FMR 16234 ^T^	–	LT906659	LT906656	–
*Neodendryphiella tarraconensis*	GZCC20_0002	–	MN999922	MN999927	–
*Periconia igniaria*	CBS 379.86	–	LC014585	AB807566	AB808542
*Periconia igniaria*	CBS 845.96	–	LC014586	AB807567	AB808543
*Pseudocoleophoma bauhiniae*	MFLUCC 17-2586 ^T^	MK347844	MK347736	MK347953	MK360076
*Pseudocoleophoma bauhiniae*	MFLUCC 17-2280	MK347843	MK347735	MK347952	MK360075
*Pseudocoleophoma flavescen*	CBS_178.93	GU238216	–	GU238075	–
*Pseudocoleophoma rusci*	MFLUCC 16-1444 ^T^	MT214983	MT185549	MT183514	–
*Pseudocoleophoma zingiberacearum*	NCYUCC 19-0052 ^T^	–	MN615939	MN616753	MN629281
*Pseudocoleophoma zingiberacearum*	NCYUCC 19-0053	–	MN615940	MN616754	MN629282
*Pseudocoleophoma calamagrostidis*	KT 3284 ^T^	LC014604	LC014592	LC014609	LC014614
*Pseudocoleophoma polygonicola*	KT 731 ^T^	AB797256	AB809634	AB807546	AB808522
*Pseudocoleophoma typhicola*	MFLUCC 16-0123 ^T^	–	KX576655	KX576656	–
*Pseudoconiothyrium broussonetiae*	CBS 145036 ^T^	–	MK442618	MK442554	–
*Pseudocyclothyriella clematidis*	MFLUCC 17-2177 ^T^	–	MT310596	MT214549	MT394730
*Pseudocyclothyriella clematidis*	MFLUCC 17-2177A ^T^	MT226667	MT310595	MT214548	–
*Pseudodictyosporium elegans*	CBS 688.93 ^T^	DQ018084	DQ018099	DQ018106	–
*Pseudodictyosporium indicum*	CBS 471.95	–	DQ018097	–	–
*Pseudodictyosporium thailandica*	MFLUCC 16-0029 ^T^	KX259524	KX259520	KX259522	KX259526
*Pseudodictyosporium wauense*	NBRC 30078 ^T^	DQ018083	DQ018098	DQ018105	–
*Pseudodictyosporium wauense*	DLUCC 0801	–	MF948622	MF948630	MF953165
*Verrucoccum coppinsii*	E00814291 conidioma ^T^	MT918778	MT918784	MT918770	–
*Verrucoccum coppinsii*	E00814291 ascoma ^T^	MT918777	MT918785	MT918769	–
*Verrucoccum spribillei*	SPO2343 ^T^	MT918773	MT918780	MT918765	–
*Verrucoccum spribillei*	SPO1154 ^T^	MT918772	MT918781	MT918764	–
*Vikalpa australiensis*	HKUCC 8797	–	DQ018092	–	–
*Vikalpa grandispora* *	KUNCC 22-12425 ^T^	OP526628	OP526638	OP526648	OP542240
*Vikalpa sphaerica* *	CGMCC3.20682 ^T^	OP526629	OP526639	OP526649	OP542241

^1^ The newly generated sequences show “*” after the species name; ^2^ type specimens/ex-type strains show “T” after the number; ^3^ missing sequences are indicated with “–”.

**Table 2 jof-08-01200-t002:** Comparison of conidia characteristics and habitats of *Dictyocheirospora*, *Digitodesmium* and *Vikalpa* species.

Species	Conidia Size (μm)	Conidia Septate	No. of Rows	Appendages	Habitat	References
*D. aquadulcis*	60–80 × 17–29	Euseptate	7	No appendages	Freshwater	[35]
*D. aquatica*	34–42 × 12.5–19.5	Euseptate	8–10	No appendages	Freshwater	[40]
*D. bannica*	73–86(–90) × 21–26(–31)	-	(5–)–7	No appendages	Terrestrial	[5]
*D. cheirospora*	54–63 × 15–26	-	5–7	No appendages	Terrestrial	[34]
*D. clematidis*	42–60 × 15–30	Distosepta	6–7	No appendages	Terrestrial	[37]
*D. chiangmaiensis*	(40–)42–46 × 16–18	Euseptate	4–6	No appendages	Freshwater	This study
*D. suae*	(65–)72–79 × (–17)20–25(–29)	Euseptate		Appendages	Freshwater	This study
*D. garethjonesii*	45.5–54.5 × 15.5–24.5	Euseptate	6–7	No appendages	Freshwater	[40]
*D. gigantica*	105–121 × 25–32	-	7	No appendages	Freshwater	[22]
*D. heptaspora*	50–80 × 20–30	-	7	No appendages	Terrestrial	[22]
*D. hydei*	(26–)30–33(–35) × 14–17	-	7	Supra-basal appendages	Terrestrial	[26]
*D. indica*	(33–)36–46(–48) × 13–18	-	6–7	Subapical appendages	Terrestrial	[26]
*D. lithocarpi*	35–40 × 12–18	Euseptate	6	No appendages	Terrestrial	[36]
*D. metroxylonis*	45–69 × 15–29	Distoseptate	4–6	No appendages	Terrestrial	[37]
*D. multiappendiculata*	(65–)72–79 × (–17)20–25(–29)	Euseptate	5–7	Subglobose appendages	Freshwater	This study
*D. musae*	45–65 × 20–27	-	7	Appendages	Terrestrial	[73]
*D. nabanheensis*	35–40 × 18–21	-	6	Appendages	Terrestrial	[9]
*D. pandanicola*	60–75 × 18.5–35.5	-	5–7	No appendages	Terrestrial	[9]
*D. pseudomusae*	(58–)61–78(–81) × 19–29(–33)	-	(6–)7	Globose to subglobose appendages	Terrestrial	[19]
*D. rotunda*	42–58 × 19–38	Distoseptate	5–7	No appendages	Freshwater	[5]
*D. subramanianii*	33–42 × 16–20	-	7	-	Terrestrial	[48]
*D. taiwanense*	(72–)74–84(–86) × 16–20(–24)	-	5	No appendages	Terrestrial	[35]
*D. tetraploides*	52.5–72.5 × 18.5– 26.5	Euseptate	5	Subapical appendages	Freshwater	[45]
*D. thailandica*	42–65 × 20–45	-	6–7	No appendages	Freshwater	[38]
*D. vinaya*	58–67 × 15.5–26.5	Distoseptate	6–7	No appendages	Freshwater	[5]
*D. xishuangbannaensis*	35–50 × 17–25	-	6	No appendages	Terrestrial	[9]
*Digitodesmium aquaticum*	(39–)41–44(–46) × (17–)19–21(–22)	Euseptate	3	No appendages	Freshwater	This study
*Di. bambusicola*	24–29–32.5 × 12.5–17–23	Euseptate	3	Appendages	Freshwater	[42]
*Di. chiangmaiense*	33–42 × 15–18	Euseptate	3	No appendages	Terrestrial	[35]
*Di. elegans*	45–60 × 12–21	Euseptate	(2–)3–4(–6)	No appendages	Terrestrial	[33]
*Di. heptasporum*	50–75 × 32.5–70	Euseptate	(6)–7	No appendages	Freshwater	[45]
*Di. intermedium*	39–76 × 25–35	Euseptate	3–11	No appendages	Terrestrial	[43]
*Di. macrosporum*	130–145 × 19–26	Euseptate	5–8	No appendages	Terrestrial	[43]
*Di. polybrachiatum*	35–54 × 15–19	Euseptate	6–9	Appendages	Terrestrial	[44]
*Di. recurvum*	30–45 × 12–21	Euseptate	4–7	No appendages	Freshwater	[41]
*Di. tectonae*	28.5–41 × 14–17	-	3–6	Subglobose appendages	Terrestrial	[74]
*Vikalpa grandispora*	(33–)38–49(–53) × (9–)13–18(–21)	Euseptate	3	Subglobose appendages	Freshwater	This study
*V. australiensis*	36–43 × 11.5–12	Euseptate	3	-	Terrestrial	[48]
*V. freycinetiae*	(27–)31–43 × 10– 20	Euseptate	3	Subglobose appendages	Terrestrial	[46]
*V. lignicola*	25–40.5 × 10–18	Distoseptate	3	No appendages	Freshwater	[5]
*V. micronesiaca*	20–30 × 10–12	-	2–4	No appendages	Terrestrial	[47]
*V. sphaerica*	(23–)26–30(–34) × (11–)16–19	Euseptate	3–4	Subglobose appendages	Freshwater	This study

## Data Availability

Not applicable.

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
