# Peer review of "Novel Species and Records of Dictyosporiaceae from Freshwater Habitats in China and Thailand"

_jof, 2022, doi:10.3390/jof8111200_

Round 1

Reviewer 1 Report

The manuscript is original, well-structured and with very good pictures. Few comments are provided below for authors:

Please define better the aims of your study at the end of the introduction.

L129: please explain better this expression "capture the origin of", since it is not clear. 

L273: I suggest using the terms "reverse" and "obverse" instead of "from below" and "from above", respectively. Also in other cases in figure captions. 

Figure 3j: this combined picture is not completely clear; please to improve cleanness try to distinguish better the reverse and obverse, maybe with labels. The same with figures 5k, 6l and 11n.

L371:"new collection ": please explain better since it is not clear. Please try to use another term in all the manuscript. Did you mean "isolated species"?

636: "Polybrachiatum": in lower case letter.

In all manuscript, please, prefer the term saprotrophic instead saprophytic or saprobic.

Author Response

Dear Reviewer

Thank you for your comments on our manuscript entitled "Novel species and records of Dictyosporiaceae from freshwater habitats in China and Thailand" (Manuscript ID: jof-1989391). Those comments are all valuable and very helpful for revising and improving our paper, as well as the important guiding significance to our researches. We have studied comments carefully and have made correction which we hope meet with approval. The responds to the reviewer’s comments, kindly see the attachment.

Reviewer 2 Report

Major Comments

In general, the manuscript is interesting about the new fungal species and new records presented from freshwater vegetal material in China and Thailand. However, some minor corrections will be needed in the current version of this manuscript in order to clarify and provide a better compression of the same. In addition, the authors should revise some comments by sections of the manuscript to improve it, as follow:

Introduction

Lines 53-54 are difficult to follow and there is no connection with the next sentence. The same occurs with the next lines until line 63.

In line 74 the authors should write the genus name, if not is confusing, Dictyocheirospora or Dictyosporium?

Results

In general, several parts of this section has a missing number of culture collection or are confusing, please revise, e.g. “MFLU 22-***”(lines 220, 241,244 there are a lot). In addition, the mycobank number is missing for some new species proposed.

In the taxonomical description, when the authors perform the in vitro culture, do they have sporulation? or is only in vegetal material, if is the last one they should comment on this section or in the discussion, it is useful to know it.

In lines 347-48, the comment is repeated from the previous description, and is not necessary here.

Line 411 revises the fungus name “D. heptepora”.

In the note of Digitodesmium aquaticum, there is a phylogenetic marker that has differences from Di. Bambusicola, could be added this information.

In lines 630-631 the authors should add the information of Digitodesmium strains studied did not grow in culture media in the taxonomical description.

Discussion

Is too short, the authors could provide another interest for this group of fungi.

Tables

In table 1 two taxa did not have the GenBank accession number, Dictyosporium tubulatum and Digitodesmium aquaticum the latter has a strange culture collection number. Please revise.

Figures

In several figures appears “MFLU 22-****”, please revise.

Author Response

(The authors gave the same response as above.)
